Winter diet of Burrowing Owls in the Llano La Soledad, Galeana, Nuevo León, México

http://orcid.org/0000-0003-2740-7305 Gonzalez Rojas Jose I. 1 jose.gonzalezr@uanl.mx
Cruz Nieto Miguel Angel 2
Guzmán Velasco Antonio 1
http://orcid.org/0000-0001-8309-7618 Ruvalcaba-Ortega Irene 1
http://orcid.org/0000-0001-9451-1925 Olalla-Kerstupp Alina 1
Ruiz-Ayma Gabriel 1
1 Universidad Autónoma de Nuevo León, Facultad de Ciencias Biologicas , San Nicolas de los Garza, Nuevo Leon , México
2 Organización Vida Silvestre A.C. , San Pedro Garza García, Nuevo León , México
Kramer Donald
Electronic publication date: 2022 Apr 27
Publication date: 2022
Volume: 10
Electronic Location ID: e13324
Received 2020 Jun 18; Accepted 2022 Apr 1
Copyright: © 2022 Gonzalez Rojas et al.
Copyright year: 2022
Copyright holder: Gonzalez Rojas et al.
License: This is an open access article distributed under the terms of the Creative Commons Attribution License, which permits unrestricted use, distribution, reproduction and adaptation in any medium and for any purpose provided that it is properly attributed. For attribution, the original author(s), title, publication source (PeerJ) and either DOI or URL of the article must be cited.
License URL: https://creativecommons.org/licenses/by/4.0/

Keywords: Burrowing Owl, Grassland, Niche breadth, Winter diet, Chihuahuan Desert, Strigidae

Funding: Universidad Autonoma de Nuevo Leon This research was funded with resources from the Universidad Autonoma de Nuevo Leon through the support program for Scientific and Technological Research (PAICyT). The funders had no role in study design, data collection and analysis, decision to publish, or preparation of the manuscript.

==============================
The dietary niche breadth of the Burrowing Owl was determined (Athene cunicularia Molina, 1782) in Llano La Soledad, Galeana, Nuevo Leon in northern Mexico, by considering prey type, numerical percentage, weight, weight percentage, frequency of occurrence percentage, and IRI percentage. The study compared data from three winters (2002–2003, 2003–2004, 2004–2005) by analyzing 358 pellets, identifying 850 prey items. Invertebrates constituted 90% of prey items, which mostly included insects (85%); beetles were the most common insects found in pellets (70%). Vertebrates made up 84% of consumed weight, of which 83% were mammals. Most of the mammals were cricetid rodents (41%). Niche breadth based on the numerical and weight percentage confirmed the Burrowing Owl as a generalist species with mean values per year ranging between 0.65 and 0.82. Additionally, there was a strong association between the weight of rodent species in winter. This association was mainly driven by changes in composition and frequency of these prey species during the second winter, probably caused by high annual rainfall. The second season also showed a statistically significant narrower niche (Ro = 0.96) and the smallest overlap (0.45 vs. 0.76) among the three winters.

Introduction

North American Burrowing Owl (Athene cunicularia Molina 1782) populations are distributed from southwest Canada, through the western and central USA (although also in Florida), and Mexico. However, most northern populations migrate to the southern USA and Mexico (Marks, Canning & Mikkola, 1999). This bird is a predator of importance that is able to maintain its prey population in stable numbers (Coulombe, 1971). The Burrowing Owl is considered an opportunistic predator (Rodriguez-Estrella, 1997) with diurnal activity, hunting mainly at dawn and dusk (Coulombe, 1971). It lives in open areas like grasslands, deserts, and disturbed areas (Coulombe, 1971; Butts, 1976; Ruiz-Aymá et al., 2019). Moreover, its habitat of discontinuous vegetation with low shrubs allows high visibility for hunting, observing predators, and keeping watch over its burrow (Coulombe, 1971; Howell & Webb, 2004). The Burrowing Owl is strongly associated with Black-tailed Prairie Dogs (Cynomys ludovicianus) and Mexican Prairie Dogs (C. mexicanus) colonies in Mexico, using their burrows for protection against predators as well as for nesting (Coulombe, 1971; Butts, 1976; Ruiz-Aymá et al., 2019).

The Burrowing Owl has shown a significant negative population trend in the United States for approximately 50 years (−0.91%/year; 1966–2015; Sauer et al., 2017). A decline has been even steeper in Canada (−6.42%/year; 1966–2015; Sauer et al., 2017), where it is listed as an endangered species (Committee on the Status of Endangered Wildlife in Canada, 2006). Additionally, the Burrowing Owl is a National Bird of Conservation Concern (U.S. Fish & Wildlife Service, 2008). Simultaneously, in México it is protected under the “Special Protection” category (Secretaria de Medio Ambiente y Recursos Naturales, 2010). The current population status of the Burrowing Owl is a result of multiple threats such as habitat fragmentation, decreased prey availability, increased predation, inclement weather, vehicle strikes, environmental contaminants, and loss of burrows (Rodriguez-Estrella, 2006; Enríquez & Vazquez-Perez, 2017).

Prey availability is one of the most important natural factors limiting populations during the winter (Newton, 1998; McDonald, Korfanta & Lantz, 2004). The majority of the studies regarding the winter diet Burrowing Owl have been conducted in the United States (Texas, Nevada, California) as well as in other countries in North and South America (Littles et al., 2007; Nabte, Pardiñas & Saba, 2008; De Tommaso et al., 2009; Andrade, Nabte & Kun, 2010). In most studies, the Burrowing Owl diet consists mainly of invertebrates, small mammals, and reptiles (Plumpton & Lutz, 1993; Littles et al., 2007; De Tommaso et al., 2009). Invertebrates are consumed most frequently (Poulin, 2003), but mammals make up most of the weight (Andrade, Sauthier & Pardiñas, 2004; Littles et al., 2007; Nabte, Pardiñas & Saba, 2008; De Tommaso et al., 2009; Andrade, Nabte & Kun, 2010; Carevic, Carmona & Muñoz-Pedreros, 2013). The occurrence of insect orders is highly variable, both temporally and spatially. The beetles (Coleoptera) and crickets (Gryllidae) volume of prey ranged from 20% to 80% in the collected pellets. Conversely, mammal species, including North American Deer Mouse (Peromyscus maniculatus), Silky Pocket Mouse (Perognathus flavus), and Merriam’s Kangaroo Rat (Dipodomys merriami), represented 98% of prey counted in the collected pellets (Ross & Smith, 1970; Coulombe, 1971; Butts, 1976; Tyler, 1983; Barrows, 1989; Mills, 2016). A study in British Columbia, Canada, indicated that 56% of the prey were insects, such as earwigs and beetles (Morgan, Cannings & Guppy, 1993). The only study of the winter diet from Mexico comes from central Mexico in Guanajuato, where 78% were invertebrates (Valdez-Gómez, 2003). Weight data were more evenly distributed among Orthoptera (26.8%), Lepidoptera (20.6%), and rodents (20.9%; Valdez-Gómez et al., 2009). The breeding season diet has also been analyzed in Durango and Nuevo Leon, where insects were the most abundant prey items (67–84%); mammals represented 50% of the weight (Rodriguez-Estrella, 1997; Ruiz-Aymá et al., 2019).

Variation in the diet has been associated with prey availability, suggesting that small mammals are selected over invertebrates when their densities are sufficiently high (Silva et al., 1995). A change in prey composition has also been associated with rainfall, with more grasshoppers and some rodents (e.g., Perognathus sp., Onychomys leucogaster) consumed during dry years but more birds consumed during wet years (Conrey, 2010). The quantity and pattern of precipitation in arid and semi-arid environments can also influence the quality of the habitat as well as abundance of prey (Ernest, Brown & Parmenter, 2000; Reed, Kaufman & Sandercock, 2007; Thibault et al., 2010). It is well established that, in general, an increase in precipitation increases coverage and small mammal diversity (Ernest, Brown & Parmenter, 2000; Thibault et al., 2010).

Information on the winter diet of Burrowing Owls in Mexico is limited; so far temporal variation has not been examined. Thus, our objective was to determine the diet composition and dietary niche breadth of Burrowing Owls over three winters (2002–2003, 2003–2004, 2004–2005) in northern Mexico (Llano La Soledad, in the southern Chihuahuan Desert). Our hypotheses are (1) that the diet composition in years with high rainfall will be different than in drier years, (2) that differences in rainfall will also affect diet niche breadth.

Study area and methods

Site description

Llano la Soledad is a plains habitat located in the northeastern Mexican state of Nuevo León, municipality of Galeana, within the Grassland Priority Conservation Area “El Tokio” (CEC & TNC, 2005; Pool & Panjabi, 2011). This area is a part of the Chihuahuan Desert ecoregion (25°9′8.87″N, 101°6′8.00″W–24°18′54.12″N, 100°23′41.48″W; Fig. 1). It is a State Natural Protected Area (Diario Oficial de la Federación, 2002) internationally known for its importance for shorebird conservation (WHSRN, 2005). It is also part of an important bird area “Pradera de Tokio” (AICA-NE-36; Del Coro-Arizmendi & Marquez, 2000) that harbors vulnerable bird species both endemic and migratory. Llano La Soledad also contains the largest colony of the Mexican Prairie Dog (Treviño-Villarreal & Grant, 1998). Therefore, it represents the most extensive, continuous habitat in terms of burrows and food availability for Burrowing Owls in northeastern Mexico (Ruiz-Aymá et al., 2016). Open grasslands dominate the area with 80% bare ground and 20% plant cover (3% of grass, 17% forbs and shrubs) (Cruz-Nieto, 2006). The semi-arid climate features temperatures ranging from 6 °C to 25 °C with an annual average of 16 °C (Comisión Nacional del Agua, 2019) and average annual precipitation of 427 mm (Instituto Nacional de Estadística Geografía e Informática, 2005).

Figure 1 Location of State Natural Protected Area Llano La Soledad, Galeana, N.L., Mexico.

Pellet collection and analyses

Pellets were collected every other day at active burrows located along 20 random transects of 1 km × 200 m, representing an area of 400 ha (5% of the Natural Protected Area). We traveled the transects daily from the first week of October through the first week of March over three winters (2002–2003, 2003–2004, 2004–2005) to collect population density data.

Each pellet was analyzed and quantified according to the mentioned by Ruiz-Aymá et al. (2019). The remains were separated into parts; the most prominent structures used to identify each group were the following: elytra, heads, tarsi, mouthparts, chelae, and stingers for arthropods; bones, teeth, feathers and scales, for mammals, birds and reptiles. We counted the number of prey items of each species in each pellet. Only the most representative structures were counted among the groups to avoid over-counting prey items. For mammals, only mandibles and cranium were counted as an individual. For birds the skull, and for reptiles, the head and limbs were counted. In the case of insects, the heads (Coleoptera) or mandible as well as mouthparts (Orthoptera, arachnids) were counted as individuals. The weight of each prey species in each pellet was also estimated. For mammals, we used the median of the weight for each species to avoid overestimation (Holt & Childs, 1991). The medians were obtained from data given for Mexico by Ceballos & Oliva (2005). For reptiles, birds, and mammals, we used specimens from Herpetology, Ornithology, and Mammalogy collections at the Universidad Autónoma de Nuevo León/Facultad de Ciencias Biológicas; for insects, data reported by Olalla (2014); for spiders, median weights were obtained from live specimens of the Arachnology collection at the Facultad de Ciencias Biológicas/Universidad Autónoma de Nuevo León. Mammals were identified according to Anderson (1972) and Roest (1991), herpetofauna according to Smith & Taylor (1950) and Smith & Smith (1993); birds according to Howell & Webb (2004) and Dunn (2006); invertebrates were identified to Borror, Triplehorn & Johnson (1989). Any vertebrate prey items that could not be identified to the species level were included in the unidentified category.

The percentage of frequency of occurrence (FO%) was calculated for each taxonomic level of prey (orders, classes, genera, species) by dividing the number of pellets, in which each kind of item was found, by the total number of pellets collected. The numerical percentage (N%) was calculated by dividing the number of items in each prey category by the total number of prey items found in all pellets. In both cases, it was multiplied by 100 to convert to percentage. The weight percentage (W%) was estimated as the total weight of each prey taxon divided by the combined estimated total weight of all prey taxa, multiplied by 100. The index of relative importance was calculated as: IRI = (N% + W%) FO%, where N% = numerical percentage, W% = weight percentage, and FO% = percentage of frequency of occurrence (Martin, Twigg & Robinson, 1996; Hart, Calver & Dickman, 2002; Marti, Bechard & Jaksic, 2007; Santana et al., 2019; Muñoz-Pedreros & Rau, 2020; Rocha, Branco & Barnilli, 2021). The IRI was divided by the total IRI, then multiplied by 100 to obtain the percent IRI (IRI%).

All protocols were performed according to the guidelines adopted by the ethics committee of the Facultad de Ciencias Biológicas of the Universidad Autónoma de Nuevo León. However, to comply with Mexican regulations, we obtained a permit (SGPA/DGVS/01588/10) granted by the Secretaría del Medio Ambiente y Recursos Naturales/Subsecretaría de Gestión para la Protección Ambiental/Dirección General de Vida Silvestre.

Statistical analyses

For each winter season, we estimated niche breadth with 95% confidence intervals (CIs) using Smith’s measure (FT) (Smith, 1982), considering years to be statistically different when the 95% CIs did not overlap. This measure considers the availability of the resource and varies from 0 (minimal) to 1 (maximal) and is therefore a standardized measure; it is a convenient measure to be used because its sampling distribution is known (Smith, 1982). A species with wide niche breadth is a generalist, while a species with a narrow niche breadth is a specialist. In addition, the overlap index of Horn (Ro) (1966) was calculated for the numerical and weight percentage using the Ecological Methodology software 7.2 (Krebs, 2011). This index varies from 0 (no common resources) to 1 (complete overlap).

To test for an association between years and the diet composition, we used χ2 contingency tests (Zar, 2010) followed by Cramer’s phi coefficient (ϕc, Cohen, 1988) as a measure of effect size, where values ϕc ≤ 0.20 represented a weak association; 0.20 < ϕc ≤ 0.60 a moderate association, and ϕc > 0.60 a strong association. For years, we used annual rainfall (Meteorological station in La Carbonera; 19032; Comisión Nacional del Agua, 2019). For the diet, the number of items and weight were used for classes, orders, and families of vertebrates and invertebrates, as well as the number of items and weight for rodent species. These analyses were conducted using PAST 4.04 (Hammer, Harper & Ryan, 2001).

Results

During the three winters, we counted an average of 11 Burrowing Owls per winter with a total of 34 and collected and analyzed 358 pellets. From the pellets, 850 prey items from 26 taxa were identified. The identified prey items represented seven orders, 17 families of invertebrates, six genera of small mammals, two genera of reptiles, and one avian genus. Vertebrates accounted for 10% and invertebrates for 90% of the total number of prey items, whereas weight percentage vertebrates accounted for 84% and invertebrates for 16%. Rodents, particularly cricetids, comprised 2% of all prey items eaten but 41% of the weight.

Insects, primarily from the orders Coleoptera (IRI% = 40; N% = 56%), and Orthoptera (IRI% = 16; N% = 27%) (Table 1), represented 82% of consumed items but contributed only 11% of the weight.

Table 1 Analysis of the winter diet of the Burrowing Owl in Llano La Soledad, Galeana, Nuevo Leon, Mexico.

	2002–2003	2003–2004	2004–2005	Total	
Prey Items	(n = 125)	(n = 116)	(n = 117)	(n = 358)	
I	N%	W	W%	P	FO%	IRI	I	N%	W	W%	P	FO%	IRI	I	N%	W	W%	P	FO%	IRI	I	N%	W	W%	P	FO%	IRI	IRI%	
Vertebrates	21	6.93	220.61	67.58	15	12.00	894.13	30	11.90	830.70	94.06	15	12.93	1,370.23	32	10.85	359	77.56	11	9.40	831.18	83	9.76	1,410.31	84.11	41	11.45	1,075.10	10.29	
Mammalia	16	5.28	201	61.58	10	8.00	534.88	17	6.75	827	93.64	11	9.48	951.94	26	8.81	359	77.56	11	9.40	812.06	59	6.94	1,387	82.72	31	8.66	776.40	7.43	
Cricetidae	4	1.32	66	20.22	2	1.60	34.46	6	2.38	482	54.58	4	3.45	196.42	11	3.73	152	32.73	6	5.13	186.97	21	2.47	700	41.71	12	3.35	148.09	1.42	
Deer Mouse																														
(Peromyscus maniculatus)	2	0.66	45	13.79	2	1.60	23.12	3	1.19	67.50	7.64	1	0.86	7.61	3	1.02	67.50	14.58	2	1.71	26.66	8	0.94	180	10.73	5	1.40	16.30	0.16	
Western Harvest Mouse																														
(Reithrodontomys megalotis)	2	0.66	21	6.43	2	1.60	11.34	1	0.40	10.50	1.19	1	0.86	1.37	8	2.71	84	18.15	4	3.42	163.70	11	1.29	116	6.89	7	1.96	16.00	0.15	
Mexican Woodrat																														
(Neotoma mexicana)	–	–		–	–	–	–	2	0.79	404	45.74	2	1.72	80.23	–	–	–	–	–	–	–	2	0.24	404	24.09	2	0.56	16.59	0.13	
Heteromyidae	4	1.32	135	41.36	3	2.40	102.43	4	1.59	65	7.36	4	2.59	23.14	9	3.05	208	44.83	4	3.42	163.70	17	2	408	24.31	11	3.07	80.84	0.77	
Merriam’s Kangaroo Rat																														
(Dipodomys merriami)	3	0.99	127.50	39.06	3	2.40	96.12	1	0.40	42.50	4.81	1	0.86	4.49	4	1.36	170	36.73	2	1.71	65.10	8	0.94	340	20.28	6	1.68	35.57	0.34	
Silky Pocket Mouse																														
(Perognathus flavus)	1	0.33	7.50	2.30	1	0.80	2.10	3	1.19	22.50	2.55	2	1.72	6.45	5	1.69	37.50	8.10	2	1.71	16.74	9	1.06	68	4.03	5	1.40	7.11	0.07	
Sciuridae	–	–	–	–	–	–	–	2	0.79	280	31.70	1	0.86	28.01	–	–	–	–	–	–	–	2	0.24	280	16.70	1	0.28	4.73	0.05	
Spotted Ground Squirrel																														
(Spermophilus spilosoma)	–	–	–	–	–	–	–	2	0.79	280	31.70	1	0.86	28.01	–	–	–	–	–	–	–	2	0.24	280	16.70	1	0.28	4.13	0.05	
Unidentified rodents	8	2.64	–	–	8	6.40	–	5	1.98	–	–	2	1.72	–	6	2.03	–	–	1	0.85	–	19	2.24	–	–	11	3.07	–	–	
Aves	4	1.32	12	3.68	4	3.20	16.00	11	4.37	–	–	3	2.59	–	6	2.03	–	–	1	0.85	–	21	2.47	12	0.72	8	2.23	7.13	0.07	
Emberizidae	1	0.33	12	3.68	1	0.80	3.21	–	–	–	–	–	–	–	–	–	–	–	–	–	–	–	–	–	–	–	–	–	–	
Black-throated Sparrow																														
(Amphispiza bilineata)	1	0.33	12	3.68	1	0.80	3.21	–	–	–	–	–	–	–	–	–	–	–	–	–	–	1	0.12	12	0.72	1	0.28	0.23	<0.01	
Unidentified birds	3	0.99	–	–	3	2.40	–	11	4.37	–	–	3	2.59	–	6	2.03	–	–	1	0.85	–	20	2.35	–	–	7	1.96	–	–	
Reptilia	1	0.33	7.61	2.33	1	0.80	2.13	2	0.79	3.70	0.42	2	1.86	1.05	–	–	–	–	–	–	–	3	0.35	11.31	0.67	3	0.84	0.86	<0.01	
Phrynosomatidae	1	0.33	7.61	2.33	1	0.80	2.13	–	–	–	–	–	–	–	–	–	–	–	–	–	–	1	0.12	7.61	0.45	1	0.28	0.16	<0.01	
Lesser Earless Lizard																														
(Holbrookia maculata)	1	0.33	7.61	2.33	1	0.80	2.13	–	–	–	–	–	–	–	–	–	–	–	–	–	–	1	0.12	7.61	0.45	1	0.28	0.16	<0.01	
Teiidae	–	–	–	–	–	–	–	1	0.40	3.70	0.42	1	0.86	0.70	–	–	–	–	–	–	–	1	0.12	3.70	0.22	1	0.28	0.09	<0.01	
Little Striped Whiptail																														
(Aspidoscelis inornata)	–	–	–	–	–	–	–	1	0.40	3.70	0.42	1	0.86	0.70	–	–	–	–	–	–	–	1	0.12	3.70	0.22	1	0.28	0.09	<0.01	
Unidentified reptiles	–	–	–	–	–	–	–	1	0.40	–	–	1	0.86	–	–	–	–	–	–	–	–	1	0.12	–	–	1	0.28	–	–	
Invertebrates	282	93.07	105.81	32.42	110	88.00	11,043.06	222	88.10	52.47	5.94	101	87.07	8,187.73	263	89.15	103.84	22.43	106	90.60	10,109.28	767	90.24	266.56	15.81	317	88.55	9,390.13	89.86	
Insecta	253	83.50	68.64	21.03	106	84.80	8,864.00	210	83.33	47.63	5.39	95	81.90	7,266.30	257	87.12	81.67	17.64	104	88.89	9,311.97	720	84.71	202.38	12.05	305	85.20	8,243.25	78.89	
Coleoptera (Beetles)	168	55.45	19.90	6.10	94	75.20	4,628.22	152	60.32	21.00	2.38	77	66.38	4,161.81	153	51.86	30.33	6.55	81	69.23	4,044.07	473	55.65	72.83	4.33	252	70.39	4,221.85	40.40	
Elateridae	–	–	–	–	–	–	–	2	0.79	0.18	0.02	1	0.86	0.70	6	2.03	0.54	0.12	6	5.13	11.05	8	0.94	0.72	0.04	7	1.96	1.92	0.02	
Carabidea	86	28.38	7.10	2.18	83	66.40	2,029.37	65	25.79	6.50	0.74	58	50.00	1,326.68	74	25.08	7.40	1.60	63	53.85	1,436.87	225	26.47	22.50	1.34	204	56.98	1,584.74	15.17	
Scarabaeidae	49	16.17	4.80	1.47	49	39.20	691.55	45	17.86	4.50	0.51	39	33.62	617.52	4	1.36	0.40	0.09	3	2.56	3.71	98	11.53	9.80	0.58	91	25.42	307.81	2.95	
Curculionidae	19	6.27	3.80	1.16	18	14.40	107.00	17	6.75	3.40	0.38	15	12.93	92.15	17	5.76	3.40	0.73	16	13.68	88.79	53	6.24	10.60	0.63	49	13.69	93.97	0.90	
Cerambycidae	14	4.62	4.20	1.29	14	11.20	66.20	7	2.78	2.10	0.24	7	6.03	18.21	23	7.80	6.90	1.49	19	16.24	150.81	44	5.18	13.20	0.79	40	11.17	66.66	0.64	
Passalidae	–	–	–	–	–	–	–	16	6.35	4.32	0.49	14	12.07	82.54	6	2.03	1.62	0.35	4	3.42	8.15	22	2.59	5.94	0.35	18	5.03	17.77	0.14	
Buprestidae	–	–	–	–	–	–	–	–	–	–	–	–	–	–	2	0.68	1.04	0.22	1	0.85	0.77	2	0.24	1.04	0.06	1	0.28	0.08	<0.01	
Tenebrionidae	–	–	–	–	–	–	–	–	–	–	–	–	–	–	21	7.12	9.03	1.95	14	11.97	180.51	21	2.47	9.03	0.54	14	3.91	11.77	0.11	
Orthoptera (Grasshoppers, crickets and bush–crickets)	73	24.09	40.38	12.37	68	54.40	1,983.56	53	21.03	26.61	3.01	43	37.07	891.20	101	34.24	50.53	10.92	68	58.12	2,624.53	227	26.71	120.37	7.18	179	50.00	1,694.29	16.21	
Acrididae	70	23.10	39.33	12.05	65	52.00	1,827.92	47	18.65	24.51	2.78	37	31.90	683.57	69	23.39	39.33	8.50	53	45.30	1,444.58	186	21.88	106.02	6.32	155	43.30	1,221.05	11.69	
Gryllidae	3	0.99	1.05	0.32	3	2.40	3.14	6	2.38	2.10	0.24	6	5.17	13.56	32	10.85	11.20	2.42	28	23.93	317.51	41	4.82	14.35	0.86	37	10.34	58.74	0.56	
Hymenoptera (Ants, bees and wasps)	2	0.66	0.36	0.11	2	1.60	1.23	5	1.98	0.02	<0.01	2	1.72	3.42	2	0.68	0.01	<0.01	1	0.85	0.58	9	1.06	0.38	0.02	5	1.40	1.51	0.01	
Vespidae	2	0.66	0.36	0.11	2	1.60	1.23	–	–	–	–	–	–	–	–	–	–	–	–	–	–	2	0.24	0.36	0.02	2	0.56	0.14	<0.01	
Formicidae	–	–	–	–	–	–	–	5	1.98	0.02	<0.01	2	1.72	3.42	2	0.68	0.01	<0.01	1	0.85	0.58	7	0.82	0.02	<0.01	3	0.84	0.69	0.01	
Dermaptera (Earwigs)	10	3.30	8	2.45	4	3.20	18.40	–	–	–	–	–	–	–	1	0.34	0.80	0.17	1	0.85	0.38	11	1.29	8.80	0.52	5	1.40	2.53	0.02	
Forficulidae	10	3.30	8	2.45	4	3.20	18.40	–	–	–	–	–	–	–	1	0.34	0.80	0.17	1	0.85	0.38	11	1.29	8.80	0.52	5	1.40	2.53	0.02	
Arachnida	29	9.57	37.17	11.40	19	15.20	318.76	12	4.76	4.84	0.55	10	8.62	45.79	6	2.03	22.17	4.79	3	2.56	2.56	47	5.53	64.18	3.76	32	8.94	83.03	0.79	
Araneae (Spiders)	25	8.25	35.09	10.75	15	12.00	228.01	4	1.59	0.68	0.08	4	3.45	5.75	3	1.02	15.93	3.44	1	0.85	3.81	32	3.76	51.70	3.08	20	5.59	38.24	0.37	
Theraphosidae	6	1.98	31.86	9.76	4	3.20	37.57	–	–	–	–	–	–	–	3	1.02	15.93	3.44	1	0.85	3.81	9	1.06	47.79	2.85	5	1.40	5.46	0.05	
Araneidae	19	6.27	3.23	0.99	11	8.80	63.89	4	1.59	0.68	0.08	4	3.45	5.75	–	–	–	–	–	–	–	23	2.71	3.91	0.23	15	4.19	12.30	0.12	
Solfugae	4	1.32	2.08	0.64	4	3.20	6.27	8	3.17	4.16	0.47	6	5.17	18.85	1	0.34	0.52	0.11	1	0.85	0.38	13	1.53	6.76	0.40	11	3.07	5.74	0.05	
Eremobatidae	4	1.32	2.08	0.64	4	3.20	6.27	8	3.17	4.16	0.47	6	5.17	18.85	1	0.34	0.52	0.11	1	0.85	0.38	13	1.53	6.76	0.40	11	3.07	5.74	0.05	
Uropygi (Whipscorpions or vinegaroons)	–	–	–	–	–	–	–	–	–	–	–	–	–	–	2	0.68	5.72	1.24	2	1.71	3.28	2	0.24	5.72	0.34	2	0.56	0.32	<0.01	
Thelyphonidae	–	–	–	–	–	–	–	–	–	–	–	–	–	–	2	0.68	5.72	1.24	2	1.71	3.28	2	0.24	5.72	0.34	2	0.56	0.32	<0.01	
Total vertebrates	21	6.93	220.61	67.58	15	12.00	894.13	30	11.90	830.70	94.06	15	12.93	1,370.23	32	10.85	359	77.56	11	9.40	831.18	83	9.76	1,410.31	84.11	41	11.45	1,075.10	10.29	
Total invertebrates	282	93.07	105.81	32.42	110	88.00	11,043.06	222	88.10	52.47	5.94	101	87.07	8,187.73	263	89.15	103.84	22.43	106	90.60	10,109.28	767	90.24	266.56	15.81	317	88.55	9,390.13	89.86	
Total	303	100	326.42	100	125	100	11,937.19	252	100	883.17	100	116	100	9,557.96	295	100	462.84	100	117	100	10,940.46	850	100	1,676.87	100	358	100	10,465.23	100	
Note:

For each taxonomic group in each of three winters and all years combined, the table shows the total number of pellets (n), number of items (I), numerical percentage (N%), weight (W), weight percentage (W%), number of pellets in which taxonomic group was present (P), frequency of occurrence percentage (FO%), index of relative importance (IRI), and percentage IRI (IRI%).

Smith’s measure of niche breadth was wide, corresponding to a generalist species. These values are consistent for the three winters, so there is no statistically significant difference in the numerical percentage between winters (2002–2003: FT = 0.79, 95% CI [0.73–0.84]; 2003–2004: FT = 0.81, 95% CI [0.76–0.86]; 2004–2005: FT = 0.82, 95% CI [0.77–0.87]; all three winters combined: 2002–2005: FT = 0.77, 95% CI [0.73–0.80]; Fig. 2A). On the other hand, the niche breadth based on the weight percentage was lower for the second winter (2003–2004: FT = 0.65, 95% CI [0.60–0.69]; Fig. 2B) than for the other 2 years (2002–2003: FT = 0.81, 95% CI [0.77–0.84]; 2004–2005: FT = 0.73, 95% CI [0.69–0.76]; all three winters combined: 2002–2005: FT = 0.71; 95% CI [0.70–0.63]; Fig. 2B). The decrease in niche widht in the second winter coincided with the precipitation of 505 mm in 2003, above the long-term average (396 mm, 1956–2014), greater than the other two winters (2002: 288 mm; 2004: 304 mm).

Figure 2 Dietary niche breadth estimate (Smith, 1982) and 95% CI for Burrowing Owls (Athene cunicularia) during three winter seasons (2002–2005) considered separately and combined based on (A) numerical percentage and (B) weight.

Horn’s index showed greater overlap in the numerical percentage between the first and the second winters (Ro = 0.96) than the overlap found between the first and third winter (Ro = 0.86) and between the second and third winter (Ro = 0.83). Regarding the weight percentage, this index showed greater overlap between the first and third winter (Ro = 0.76), than between the first and second winter (Ro = 0.45), and the second and third winter (Ro = 0.45).

Based on the ϕc values, there was a weak association between the winters and the number of items for vertebrates classes (χ2 = 5.82, df = 4, p < 0.0001, ϕc = 0.18) and a moderate association for families (χ2 = 14.26, df = 10, p < 0.0001, ϕc = 0.20) and rodent species (χ2 = 15.07, df = 10, p < 0.0001, ϕc = 0.43). We found a weak association between winters and the number of elements for invertebrate classes (χ2 = 15.43, df = 2, p < 0.0001, ϕc = 0.14) and a moderate association for orders (χ2 = 65.22, df = 12, p < 0.0001, ϕc = 0.21) and families (χ2 = 221.50, df = 32, p < 0.0001, ϕc = 0.38).

In terms of weight, there was a weak association between winters and weight for vertebrates classes (χ2 = 89.09, df = 4, p < 0.0001, ϕc = 0.17), a moderate association for families (χ2 = 643.93, df = 10, p < 0.0001, ϕc = 0.47), and a strong association for rodent species (χ2 = 1,010.4, df = 10, p < 0.0001, ϕc = 0.61).

The strong association observed in the rodents species during the second winter season was due to the greater consumption of Spotted Ground Squirrel (Xerospermophilus spilosoma) and Mexican Woodrats (Neotoma mexicana), as well as decreased consumption in Merriam’s Kangaroo Rat during the same period (Table 1). There was a moderate association between winters and weight for invertebrate classes (χ2 = 14.82, df = 2, p < 0.0001, ϕc = 0.24), orders (χ2 = 58.72, df = 10, p < 0.0001, ϕc = 0.34), and families (χ2 = 97.86, df = 26, p < 0.0001, ϕc = 0.44).

Discussion

Our findings provide additional evidence that the Burrowing Owl is a generalist, opportunistic predator. Invertebrates (mainly arthropods) were the most common, abundant food items, corroborating previous studies, showing that overwintering Burrowing Owls feed mainly on arthropods and small mammals (Ross & Smith, 1970; Coulombe, 1971; Butts, 1976; Tyler, 1983; York, Rosenberg & Sturm, 2002; Valdez-Gómez, 2003; Littles et al., 2007; Hall, Greger & Rosier, 2009). Invertebrates composed 90% of the total prey items consumed, similar to other studies (Littles et al., 2007; Carevic, Carmona & Muñoz-Pedreros, 2013; Cavalli et al., 2014) that report values ranging from 93% to 98%; however, it was higher than the 78% reported by Valdez-Gómez (2003) for Mexico. Insects contributed 84% to the diet of the Burrowing Owl, which was very similar among the winters, varying between 83% and 87%. This value is greater than the 63% reported in Mexico (Valdez-Gómez, 2003) but lower than the 91% registered in southern Texas (Littles et al., 2007).

Beetles were the most frequently consumed insects (56%), with an average variation of 11% during the years considered for the study. Beetles are not frequently observed as prey in North America, and were mostly recorded during the breeding season (39–54%; Haug, 1985; Green et al., 1993; Floate et al., 2008), whereas for South America beetles are more common as prey (e.g., Andrade, Nabte & Kun, 2010; Cavalli et al., 2014). In most North American studies, crickets (Gryllidae) were the most frequently ingested insects (York, Rosenberg & Sturm, 2002; Valdez-Gómez, 2003; Littles et al., 2007; Hall, Greger & Rosier, 2009). In our study, carabid beetles were the most frequently consumed (26%), while other authors report Gryllidae (crickets; Valdez-Gómez, 2003; Littles et al., 2007). Jonas, Whiles & Carlton (2002) observed a positive correlation between native vegetation and beetles, whose consumption by Burrowing Owls in our study was likely related to the high proportion of native vegetation in Llano La Soledad. Beetles have an affinity for native vegetation (Crisp, Dickinson & Gibbs, 1998; Jonas, Whiles & Carlton, 2002; Littles et al., 2007). On the other hand, crickets are commonly in disturbed areas (Jonas, Whiles & Carlton, 2002) in North America, especially in grazed and overgrazed pastures, abandoned pastures (Jonas, Whiles & Carlton, 2002), abandoned crop fields, lawns, old fields, other grassy areas (Cade & Otte, 2000; Moulton, Brady & Belthoff, 2005), as well as in tilled and cultivated fields (Carmona, 1998); however, these types of fields were uncommon in our study area, the closest being approximately 10 km away. Conversely, in South America, although beetles are highly consumed and preferred by the Burrowing Owl, their relative abundance was higher in agricultural areas than in vegetated sand-dunes (Andrade, Nabte & Kun, 2010; Cavalli et al., 2014; Cadena-Ortiz et al., 2016). These authors suggested that beetles may also have been common in the owl diet because they require little effort to capture, particularly when they are abundant near burrows. Littles et al. (2007) reported that beetles were the second-most consumed (32%) of all prey species on a barrier island, where vast expanses of the native vegetation occur compared to agricultural areas and grasslands. The second-most frequently consumed prey species in our study were grasshoppers (27%) for the 3 years studied; Valdez-Gómez (2003) reported this same group (15%), while Littles et al. (2007) mentioned Lepidoptera (13%). When analyzing our data, a variation in the numerical percentage was observed amongthe arthropod groups, for example, the spiders presented a value of 8% in the first year, decreasing in the rest of the years. Insects, such as Scarabaeidae, decreased in the third year (1%), whereas Tenebrionidae was only present in the third winter season, while Gryllidae increased in the third winter season (11%) (Table 1). The wide variety of insect prey consumed at of Llano de la Soledad, N.L., confirms the opportunistic foraging of the Burrowing Owl; in other words, it feeds on whatever is available in a natural habitat (Jaksić & Marti, 1981; Jaksic, 1988; Green et al., 1993; Littles et al., 2007). Vertebrates contributed 10% to the diet of Burrowing Owls, which is lower than the 21% recorded in Guanajuato, Mexico (Valdez-Gómez, 2003), but greater than the 2% recorded in southern Texas (Littles et al., 2007). However, rodents were the most frequent vertebrates with 71%, similar to the 70% reported by Littles et al. (2007) and lower than 86% of Valdez-Gómez (2003).

We found that the Western Harvest Mouse was the most common rodent prey (19%), followed by the Silky Pocket Mouse (15%), Deer Mouse, and Merriam’s Kangaroo Rat (13% each). In contrast, the most commonly found rodents in Guanajuato were Deer Mouse (39%), and Silky Pocket Mouse (35%) (Valdez-Gómez, 2003); whereas in Texas, the most common were Northern Pigmy (23%), and Fulvous Harvest Mouse (19%) (Littles et al., 2007). All of these rodent species are distributed in U.S. and Mexico, mostly within arid areas of both countries, their variation as the most consumed prey per region is consistent with the capacity of the Burrowing Owl to use what is likely most available in each region. According to the IRI, invertebrates were the main food component, with insects, particularly Coleoptera and Orthoptera, being the most abundant. However, there was larger prey (vertebrates, arachnids) that were either eaten rarely or predominated in the samples because they were digested at a slower rate, as mentioned by Hart, Calver & Dickman (2002) (Table 1).

Even though vertebrates only represented 10% of total prey items, they accounted for 84% of the total weight consumed, similar to the findings of other authors (Littles et al., 2007; Nabte, Pardiñas & Saba, 2008; Carevic, Carmona & Muñoz-Pedreros, 2013). Mammal weight was 83%, varying between 62% and 93% among years, which is higher than what has been reported for Texas (52%) (Littles et al., 2007) and Mexico (25%; Valdez-Gómez et al., 2009), but within the 25–95% reported in Argentina, and Chile (Andrade, Sauthier & Pardiñas, 2004; Nabte, Pardiñas & Saba, 2008; De Tommaso et al., 2009; Andrade, Nabte & Kun, 2010; Carevic, Carmona & Muñoz-Pedreros, 2013). Cricetid rodents comprised 42% of the weight, falling within the range of 37–95% found in other studies (Littles et al., 2007; Nabte, Pardiñas & Saba, 2008; Andrade, Nabte & Kun, 2010).

Changes in rodent weight during the second winter regarding the consumption of vertebrates drove the main differences in niche breadth and prey composition among the years studied. These differences coincided with a high annual rainfall that may have resulted in irruptive population events (Greenville, Wardle & Dickman, 2012) or caused changes in population densities of rodent species, which would have affected their availability for Burrowing Owl (Silva et al., 1995; Thibault et al., 2010; Ernest, Brown & Parmenter, 2000). Although this was not measured, the temporal variation in populations of all prey taxa in our study have been associated with rainfall, more strongly for the species we found had changed the most, such as Merriam’s Kangaroo Rat, Silky Pocket Mouse, Spotted Ground Squirrel and Western Harvest Mouse (Whitford, 1976; Brown & Zeng, 1989; Brown & Ernest, 2002).

Conclusions

These results represent the first systematic effort to investigate the winter diet of Burrowing Owl in prairie dog colonies in northeastern Mexico. The southern Chihuahuan Desert, where the study was conducted, contains the largest expanse of Mexican prairie dog colonies harboring winter populations of Burrowing Owl and other birds with conservation status in North America. Temporal studies that include prey availability in disturbed and undisturbed areas of the southern Chihuahuan Desert would clarify the dynamics of prey use, as well as the of preference for this vulnerable owl species. It would also be instructive to examine the effects of variation in vertebrate weight consumption on the survival of Burrowing Owl during wet and dry years, especially considering climate change scenarios. Another relevant aspect of the temporal framework for diet studies is their relationship with pesticides and indirect exposure to contaminated prey, which is likely, although with limited evidence at the moment (Haug & Oliphant, 1990; James, Fox & Ethier, 1990).

Finally, it is also important to highlight that Llano La Soledad grasslands are key to maintaining healthy populations of the Burrowing Owl as well as other species (Golden Eagle, Long-billed Curlew, Mountain Plover, Worthen’s Sparrow). The conservation and management of this population depend on the depth of our knowledge of the natural history of this species, including its foraging ecology.

Supplemental Information

Supplemental Information 1 Raw data.

The consumption of the Burrowing Owl diet during the three winter periods. As well as the average weight of the registered preys.

Click here for additional data file.

Supplemental Information 2 Abbreviation.

Click here for additional data file.

Additional Information and Declarations

Competing Interests

Author Contributions

Field Study Permissions

Data Availability

The authors declare that they have no competing interests.

Jose I. Gonzalez Rojas conceived and designed the experiments, performed the experiments, analyzed the data, prepared figures and/or tables, authored or reviewed drafts of the paper, and approved the final draft.

Miguel Angel Cruz Nieto conceived and designed the experiments, prepared figures and/or tables, database, and approved the final draft.

Antonio Guzmán Velasco conceived and designed the experiments, prepared figures and/or tables, and approved the final draft.

Irene Ruvalcaba-Ortega conceived and designed the experiments, performed the experiments, analyzed the data, prepared figures and/or tables, and approved the final draft.

Alina Olalla-Kerstupp conceived and designed the experiments, prepared figures and/or tables, language translation, and approved the final draft.

Gabriel Ruiz-Ayma conceived and designed the experiments, performed the experiments, analyzed the data, prepared figures and/or tables, authored or reviewed drafts of the paper, and approved the final draft.

The following information was supplied relating to field study approvals (i.e., approving body and any reference numbers):

All protocols were performed according to the ethical guidelines adopted by the ethic committee of the Facultad de Ciencias Biologicas of the Universidad Autonoma de Nuevo Leon.

However in order to comply with the Mexican regulations we have a permit (SGPA/DGVS/01588/10), granted by the Secretaria del Medio Ambiente y Recursos Naturales/Subsecretaria de Gestion para la Proteccion Ambiental/Direccion General de Vida Silvestre.

The following information was supplied regarding data availability:

The raw data are available in the Supplemental File.

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
