# Peer review of "Winter diet of Burrowing Owls in the Llano La Soledad, Galeana, Nuevo León, México"

_PeerJ, doi:10.7717/peerj.13324_

## Round 0.1 · original submission · Major Revisions

As a result of the timing of acceptances to our invitations to review, we received three reviews of your manuscript. All find that the study makes a contribution to our understanding of the diet of burrowing owls, but recommend changes before publication. And all have provided thoughtful comments to improve the manuscript.

Major Concerns

Methods
Some important details of the Methods are missing.
* It is not clear how many individual owls may have produced the pellets found and what the spatial extent of the area over which they were gathered. In terms of representative data, going out to the same burrow every few days is very different from going to a completely different area to gather pellets each time.
* As the reviewers indicate, the reader needs more information on your fundamental procedures and methods beyond the references to Errington and Marti (L96-97). Clearly define the measures you derived from the data. The reference to one individual per pellet (L103) is confusing and seems critical to understanding your data.

Explanation of differences between years
With only three years, you do not have the evidence for a strong conclusion about the effect of rainfall (e.g., L220). It is appropriate to discuss rainfall as a possible cause, including processes that might be involved, but with only 1 high rainfall and 2 lower rainfall years, you do not have the evidence for a firm conclusion. You could consider the suggestion of a reviewer to subdivide your data by months. That could reveal consistency within years if the sample size is adequate, but would not resolve the fundamental problem that there is only one high-rainfall year.

Language use
* The use of English is quite good, but there are numerous small errors of style and grammar and some sentences that are unclear. I have provide an annotated pdf with corrections to those errors that I noticed. In your reply to the reviews, you do not need to mention all these small changes unless you disagree with the suggestions.
* Please avoid multiple small paragraphs as in discussion of insects in the diet. Organize the material logically into related sets of topics and introduce each paragraph with an appropriate topic sentence. I have indicated a possible structure in the section on insect diet, but the same concern applies to the mammalian components of the diet.

Raw data
Please see the PeerJ guidelines regarding raw data. I did not find the raw data in the files available.

Other Suggestions

Study species. I agree with the reviewers that a more complete introduction to your study species would be useful to readers. Consider starting the introduction with a short paragraph summarizing its distribution, habitat and foraging behavior before raising the issue of population decline.

Study site. I think you could condense the description of your study site. Consider what readers need to know to understand your article and its implications and delete the excess. In particular, lists of other bird and plant species (L77-88) present do not seem to be needed.

Tables.
*I agree with the reviewer that you should consider combining Tables 1 and 2. Avoid redundancies between tables and figures. If I understand correctly, Fig. 4 partially overlaps with Table 3. I suggest that a two-panel figure would be a clearer presentation of the data than the table. Other information in Table 3 could be added to the table heading or the text.
* I do not agree with Reviewer 1, suggestion 4, that you should attempt to put information regarding other studies into a table. That could be part of a larger comparative review but is not the objective of this study.

Figures. I agree with the reviewer that Fig. 1 and 3 are not needed.

References
Generally, the references appear to be quite carefully done. However, there are a few places where capital letters have been retained in journal article titles and italics are missing for species names.

L39 and elsewhere. Please fully indent paragraphs after the first in a subsection or skip a line so that the paragraph structure is clear.
L39-41. Sentence unclear. The sentence implies that it has been mostly studied in the U.S. and other countries of North and South America. This covers the whole range. Do you mean that the majority of studies have been in the U.S. but there have also been studies in other countries across North and South America?
L53-54. This phrase is not clear and the sentence is too long. Please clarify what it refers to and make the phrase into a separate sentence.
L103. Not clear. One individual owl or one individual prey item?
L129 and elsewhere. Define all abbreviations at first use. I suggest providing the term and abbreviation in Methods/Statistical Analyses.
L131. Finish the comparison: smaller than what?
L134. It is not clear what a correlation between yearly parameters and invertebrate classes means.
L157. Do not use ‘seasons’ to refer to differences between years. It implies a difference between winter and other times of year that you did not study.
L183-186. I do not fully understand what point you are making here. The sentence seems to be run-on from the previous one but its relationship is unclear. Delete or divide sentences and clarify.
L189-192. In the Discussion, a brief review of relevant findings from the current study is sometimes needed before relating the findings to the literature. However, such review should not be extensive. This material appears to be only results and not discussion.
L193-196. This sentence appears out of place. What is the connection to previous discussion of insects in the diet?
L219ff. Turn this section into a Conclusions section, using the present emphasis on future directions of the research and minimizing overlap with the previous discussion.

Reviewer 1 ·

Basic reporting

'no comment'

Experimental design

'no comment'

Validity of the findings

'no comment'

Additional comments

The study deals with the dietary niche breadth of burrowing owls Athene cunicularia in northern Mexico. The authors consider prey type, frequency of occurrence, and biomass during three consecutive winter periods. The sample size was adequate and authors identified more than 800 prey items. Vertebrates accounted for the majority of the consumed biomass with Cricetid rodents being the most important prey species. With regard to invertebrates, beetles were the most commonly found in pellets. Based on the figures of niche breadth the burrowing owl is regarded as a generalist species though a significant association between relative biomass of rodent species and invertebrate families was discovered. The authors justify this pattern as a result of rainfall variations. Overall I find the study interesting and meriting publication. The manuscript is well written, the sample sizes are adequate for statistical analysis and the analytical procedure is indicative for dietary studies of raptors. I have a number of general comments that could improve the manuscript and assist the reader to extract the most of its value:
1) In all tables apart from a grand total per season, relevant subtotals per taxa would be interesting to be depicted.
2) Generalist vs. specialist division with regard to winter diet in nocturnal birds of prey is very common in all food ecology studies. However I believe that many studies misjudge dietary shifts as food specialization. I wonder if the authors have data on the abundance of the potential or actual prey species on the study area. Moreover data on their availability would be mostly valuable. I would suggest to make a short paragraph on the ecology of the prey species such as their breeding cycle, or when population peaks are expected and connect them to the findings of the study.
3) In the discussion section, it would be interesting to make some comments on the owl’s feeding behaviour and hunting strategy. I wonder if rainfall have an indirect impact on the species due to the vegetation height or structure that could impede foraging.
4) Last perhaps all the data on the biomass, prey items, niche breadth, the study area and the reference from different studies could be included in a table. This would make comparisons much easier.

·

Basic reporting

The manuscript is well written, with no need for major corrections, but some sentences are a little confused.
There is a vast literature on the subject (burrowing owl diet) and, although the authors have cited many relevant papers, I think it was necessary to add some studies carried out in the Neotropical region (I suggested some to the authors).
Figures 1 and 3 do not present relevant information and can be deleted. The species is well known and images can be found easily in an internet search. Images of some remains of the prey, just as an example, do not add anything relevant to the manuscript.
Tables 1 and 2 can be merged. The information on Biomass and the frequency of ccurrence can be presented in a single table.
Table 5 can be deleted and information added to the body of the manuscript.

Experimental design

The study is basically descriptive of the species' diet, and although the authors have proposed to analyze the temporal variation and the relationship of the diet to the year rainfall at the study site, the analyzed period was too short to assess temporal patterns. Only 3 years were evaluated, with one year differing from the others. Result that can be merely casual, not reflecting a pattern in the diet, especially in the case of a generalist and opportunistic species, in which changes in the diet would be expected according to changes in prey availability.
The analisys of biomass surely undenderestimates data for invertebrates, sice the authors considered only one individual per pellet of each prey identified (Methods, ln.102-103). Also, the authors do not explain how they estimated the biomass for birds, most of which have not yet been identified at the Family level.
According to the manuscript, 358 samples (pellets) were collected, however, the authors did not reported how many nests and/or individuals the sample is representative. If the representativeness is low (1, 2, 3 couples of birds/nests) the result does not have much power to represent a pattern of the species' diet, even for the studied region.

Validity of the findings

At several parts on the discussion, the authors apparently use FO as an index of relative abundance (or relative density). For example: 1) Line 153 - “Invertebrates represented 90% of prey items consumed…”; 2) Line 157 - “Insects represented 84% of the items in the diet…”; 3) line 197 - “Vertebrates represented 10% of the remaining prey items consumed by the Burrowing Owls…”. The study presents Frequency of Occurrence data, therefore, the percentage is an indication of how commonly a certain prey is consumed by the owl, but it does not mean that it is greater or less representative in terms of the quantity of items consumed, as suggested by the sentences above.

Additional comments

Dear authors, I would like to congratulate you for the study, which brings new information about Bowl's diet during the non-reproductive season, a period in which we find less information in the literature about the species' diet. The manuscript is well written, with some confusing sentences that could be rewritten. However, there are some issues that I would like to discuss with you, which I think weaken the study, but which can be easily resolved.
The first issue is the temporal assessment of the diet. Although the results showed a relationship between diet and rainfall, the study covered only 3 years, a short time to establish a pattern of diet variation. Since the pellets where collected every 3 days, perhaps refining the analysis for a monthly assessment would reinforce the results found for the interannual analysis. I think the result presented is relevant, but as it is one of the main objectives of the work (Introduction, ln. 66-68), it weakens the study as a whole. I suggest maintaining the description of the Burrowing owl diet as the main objective of the study, presenting the temporal analysis and relationship with rainfall, but maintaining caution when analyzing these data.
Regarding the analysis of the diet, my main criticism is in the calculation of biomass assuming that there was only one individual per sample (Methods, p. Ln.102-103). Would not invertebrates be totally underestimated in this analysis? Although invertebrates tend to have less representativeness in terms of biomass, they are generally numerically higher (relative density), and the absence of relative density data (e.g., minimum number of items consumed) skews the result. It is relatively easy to estimate the minimum number of individuals consumed based on single structures (skulls, heads, pronotes), paired structures (elytra, jaws) or multiple (limbs, wings) and would add more information to the diet description giving a result more robust for estimating biomass and relative importance of each type of prey in the Burrowing owls diet. The use of Frequency of Occurrence (FO) alone to analyze the diet is also complex, especially when comparing the results with to other studies. On the one hand, FO reduces the effect of opportunistic consumption of aggregate prey (e.g., ants and termites), on the other, it gives equal weight to prey that is more abundant and to those that are rarer. If both type of prey (rare and abundant) are consumed regularly (eg, FO> 70%), but according to their availability, a prey that is little consumed because it is scarce (eg, relative density = 0.5 ind./pellet), but regularly consumed ( eg, FO = 85%), will appear to be as relevant as abundant prey (eg, relative density = 20 ind./pellet) also consumed regularly (eg, FO = 90%). My suggestion would be to include information on the relative density (RD) of individuals and calculate a composite index such as the IRI (Pinkas et al. 1971. Calif. Fish Game 152: 1-105), which aim to equalize FO, RD and Biomass. Compound indices can be problematic, but when presented together with FO, RD and biomass, they can be of great value (e.g., Cortés 1998. Canadian Journal of Fisheries and Aquatic Sciences 55 (12): 2708).
Still in relation to biomass, it is not explained in the manuscript how the mass was estimated for the birds, most of which have not yet been identified at the Family level. This information is highlighted at the end of the results (lines 145 and 146), so the way in which the birds' biomass was estimated should be reported.
Finally, I would like to know how many individuals or nests were included in the analysis. You collected 358 samples (pellets), however, more important than informing the total samples collected is informing how many nests and/or individuals the sample is representative, as this affects the amplitude of the results. If the representativeness is low (1, 2, 3 couples of birds/nests) the result does not have much power to represent a pattern of the species' diet, even for the studied region. Especially being one of the regions with potentially high density of Burrowing owls (according to the authors themselves, the region "... the most extensive and continuous habitat in terms of burrow and food availability for the Burrowing Owl in northeastern Mexico".).
Below I present some specific questions that need clarification or that can be improved in the manuscript.

1) Introduction
There is a vast literature on Burrowing owl, especially on diet, many of which were cited in the manuscript, however, I would like to suggest some inclusions.
Lines 34 -37 - Several causes related to the decline of the owl populations in the Northern Hemisphere are listed, however based on data from Canada (Environment Canada, 2012). I’d like to suggest consulting Enríquez and Peréz (2017 - Los Búhos de México - In Enríquez (ed). Neotropical Owls: Diversity and Conservation) who on page 487-490 comments on the effect of pesticides on eggs (Garcia-Hernández et al. 2006); on illegal trade (Sosa-Escalante 2011); and on loss of habitat and other threats (Chávez-Ramírez 1990. Rodríguez-Estrella y Granados 2006, GarcíaHernández et al. 2006). I also suggest citing Rodriguez-Estrella and Granados (2006 - page 191, Table 3; In Rodriguez-Estrella (ed) Current Raptor Studies in Mexico).
Lines 38-39 - Although “Food availability is one of the most important natural limiting factors in populations during winter (Newton, 1998; McDonald et al., 2004).”, I don’t thing that is the case to Burrowing owl. It’s an opportunist and generalist species that will basically changes from one type of prey to another according to their availability. Also, you didn’t evaluated the food availability, so I don’t think that this sentence is related to the study or justifies the study as it seems the authors intended to including it in the Introduction.
Lines 40-42 - I suggest including more references on the species' diet in addition to Plumpton and Lutz (1993) and Poulin (2003). When writing “The winter diet of the Burrowing owl has been studied in the United States in Texas, Nevada, and California, as well as in other countries in North, and South America and consists of invertebrates, small mammals, and reptiles (Plumpton & Lutz, 1993). ” you suggests that information about the diet of Burrowing owls for these regions is scarce, when in fact it is vast. While “Invertebrates are consumed most frequently (Poulin, 2003) but mammals compose most of the biomass (Andrade et al., 2004; Littles et al., 2007; Nabte et al., 2008; De Tomasso et al., 2009; Andrade et al., 2010; Carevic et al., 2013). ”, suggests that only Poulin (2003) identified higher consumption of invertebrates, while many more studies found higher consumption of vertebrates, which is not completely true. Vertebrates may be more relevant in terms of biomass, but invertebrates are more representative numerically (in the discussion you highlight that most of prey eaten by the Burrowing owl is invertebrates). I strongly suggest that you rewrite these sentences so as not to induce the reader to a misunderstanding about the knowledge of the Burrowing owls diet.
Lines 66-68 - I wonder what your hypothesis is about the relationship between diet and rainfall. “We predicted that in a year with high rainfall, diet composition will be different than in drier year…” is a very broad hypothesis. How would you expect the diet to change in relation to local railfall? For example, what kind of prey would be most consumed in a drier year? And in the rainiest? What is the theoretical support to expect this variation? The hypothesis should be well defined and supported by the literature, which was mentioned briefly, in your Introduction, in the previous paragraph.

2) Study Area and Methods
Lines 80-83 - “The Llano La Soledad… represents the most extensive and continuous habitat in terms of burrow and food availability for the Burrowing Owl in northeastern Mexico.” - is this conclusion based on any study or is it an inference of the authors?
Lines 90-92 - “Annual rainfall for 2002-2004 years, was obtained from the closest (~ 6 km) meteorological station in La Carbonera (19032; 92 CONAGUA, 2019).” - this information is loose in this paragraph. I suggest putting in the topic “Statistical Analyzes”, explaining that an analysis of the diet was carried out with the average precipitation of the years studied.

3) Results

Tables 1 and 2 can be joined. For each item you can inform in the table cell: FO (B) (e.g., Peromyscus maniculatus 0.7 (10.7), or the reverse B (FO)).
Table 5 can be deleted and the information mentioned in the text.
Line 128 - “Niche breadth measures were wide, indicating, as expected, a generalist species,…” - I suggest removing “as expected” because it is a term that “discusses” the result.
Lines 129-130 - “… with consistent overall estimates for both frequency of occurrence (FT = 0.77; 95% CI = 0.74-0.80) and biomass (FT = 0.74; 95% CI = 0.70-0.77).” - although FT refers to Smith's measure of niche breath, it is possible that some readers are unfamiliar with this measure, so I suggest including this information in the first mention, eg ”... both frequency of occurrence (Smith's measure [FT] = 0.77; 95% CI = 0.74-0.80) and biomass (FT = 0.74; 95% CI = 0.70-0.77). ”
Lines 145-146 - as I mentioned earlier, how did you calculate the biomass of birds?
4) Discussion
At several parts on the discussion, the authors apparently use FO as an index of relative abundance (or relative density). For example: 1) Line 153 - “Invertebrates represented 90% of prey items consumed…”; 2) Line 157 - “Insects represented 84% of the items in the diet…”; 3) line 197 - “Vertebrates represented 10% of the remaining prey items consumed by the Burrowing Owls…”. The study presents Frequency of Occurrence data, therefore, the percentage is an indication of how commonly a certain prey is consumed by the owl, but it does not mean that it is greater or less representative in terms of the quantity of items consumed, as suggested by the sentences above.
Line 177 - 179 - “Conversely, in South America, although coleopterans have been found to be highly consumed and preferred by the burrowing owl, their relative abundance was higher in agricultural areas than in vegetated sand-dunes (Cavalli et al., 2013) … ”- Coleopterans are an important item in the Burrowing owl diet in South America and it is worth mentioning more studies that support this in addition to Cavalli et al (2013). As a suggestion, you could mention some of these studies:
ANDRADE, A., M.J. NABTE, AND M.E. KUN. 2010. Diet of the Burrowing Owl (Athene cunicularia) and its seasonal variation in Patagonian steppes: implications for biodiversity assessments in the Somuncurá Plateau Protected Area, Argentina. Studies on Neotropical Fauna and Environment 45: 101–110.
CADENA-ORTIZ, H.F., C. GARZÓN, S. VILLAMARÍN-CORTÉZ, G.M. POZO-ZAMORA, G. ECHEVERRÍA-VACA, J. YÁNEZ, AND J. BRITO M. 2016. Diet of the Burrowing Owl Athene cunicularia, in two locations of the inter-Andean valley Ecuador. Revista Brasileira de Ornitologia 24(2): 122-128.
CAREVIC, F.S, E.F CARMONA, AND A. MUÑOZ-PEDREROS. 2013. Seasonal diet of the burrowing owl Athene cunicularia Molina, 1782 (Strigidae) in a hyperarid ecosystem of the Atacama desert in northern Chile. Journal of Arid Environment 97: 237-241
DELGADO-V., C.A. 2007. La dieta del Currucutú Megascops choliba (Strigidae) en la ciudad de Medellín, Colombia. Boletín SAO XVII: 111-114.
MOTTA-JUNIOR, J.C.. 2006. Relações tróficas entre cinco Strigiformes simpátricas na região central do Estado de São Paulo, Brasil. Revista Brasileira de Ornitologia 14: 359-377.
NABTE, M.J., U.J.F. PARDIÑAS, AND S.L. SABA. 2008. The diet of the Burrowing Owl, Athene cunicularia, in the arid lands of northeastern Patagonia, Argentina. Journal of Arid Environments 72: 1526–1530
TOMMASO, D.C., R.G. CALLICÓ FORTUNATO, P. TETA, AND J.A. PEREIRA. 2009. Dieta de la Lechucita Vizcachera (Athene cunicularia) en dos áreas con diferente uso de la tierra en el centro-sur de la provincia de La Pampa, Argentina. Hornero 24: 87-93

·

Basic reporting

1.) Although not bad in general, a careful review of the entire manuscript is needed in order to correct the numerous English errors (i.e., sentence structure, spelling, etc.).

2.) I’m comfortable with the Introduction and background as it relates to the burrowing owl (BUOW) diet literature. However, as this manuscript is intended as a full scientific paper, I’d suggest a brief description of the owl, and perhaps a description of its distribution and habitat preferences.

3.) Conformation of the manuscript structure with peerj standards seems good. However, placing a comma after the author’s name and before the parentheses, in a citation (e.g., Smith, (2015) is not correct format. I’d suggest removing these commas.

4.) Figures are acceptable in general, although captions, which belong below the figure, should be edited for clarity, and then shortened. Specifically, for Fig. 1, I’d suggest cutting out the term “study species”, Fig. 2 seems fine, Fig. 3 is good, and Fig. 4 is ok, but the axes must be labeled.

5.) The Literature Cited section is extensive, which is usually a very good thing. However, perhaps a reduction in the number of references might be warranted here.

6.) More reference to tables and figures in the Discussion would help reduce the amount of text.

7.) Some of the terminology used for diet parameters is confusing, and therefore, because of its relative importance, I will mention this issue again. As an example, on line 44 of the PDF version of the manuscript, as well as in Table 1, the authors use the term “frequency of occurrence”. Frequency data (not presented) refer to the number of individual prey items in a sample from that prey taxon. Frequency values would be whole numbers.

Table 1 data are presented as percentages. I believe these values to actually be “Percent of Total Prey Items”. The authors should review these data. Then, a “Frequency” column and perhaps even a “Percent Occurrence” column could be added to Table 1. Percent Occurrence shows the percentage of all pellets containing that prey item. All tables should be carefully reviewed by the authors.

Experimental design

1.) Primary research seems to be within the scope of this journal.

2.) The specific research questions and hypotheses should be clarified.

3.) The results from this study seem to fill a regional knowledge gap.

4.) Description of study area is good.

5.) Following pellet collection, all information relative to diet analysis, such as pellet dissection, prey identification, and methods of quantification are left to the literature. At least a brief summary of techniques used during this process is critical.

6.) Seems that prey identifications were primarily made using taxonomic keys, or plates in field guides and other references. Making Species and sometimes even Genus-level IDs using the literature can be challenging at best. The best way to make such IDs requires the use of comparative Osteological and Invertebrate collections. I see no evidence that the authors did this.

7.) Quantification methods for the diet analysis must be clarified. The authors should present data as Frequency, Percent Occurrence, Percent of Total Prey Items, etc. Check the literature for published examples of how other authors have presented their diet data.

8.) For any tabular listing of prey taxa, those taxa need to be presented in proper phylogenetic order. For example in Table 1, Peromyscus, Reithrodontomys, and Neotoma are Genera within the same Family Cricetidae. Two Heteromyids and a Sciurid separate these taxa at this time. Please make these changes. Also, tables should also include common names of all prey taxa that have them.

Validity of the findings

1.) The primary diet data have been presented, or they are at least available to the authors. For example, I’d suggest that Frequency data for each prey taxon should be included. Also, remember that without the use of comparative collections, species-level identifications of osteological (skeletal) remains will always be in question.

2.) Conclusions seem sound.

3.) Once again, I believe that the results of this study, once the manuscript is edited for errors, and once additional prey parameters are included, represent a significant addition to the knowledge of the BUOW winter diet.

Additional comments

No Comments

---

## Round 0.2 · Minor Revisions

Two of the first reviewers were available to provide feedback on your revised manuscript. Reviewer 1 indicated that the changes were appropriate and that your manuscript could be published in its current form. Reviewer 2 agrees that you have successfully answered the concerns raised earlier. However, this reviewer also noted two remaining issues, the definition of Frequency of Occurrence and the completeness of Table 1 and provided numerous suggestions for clarification of the writing. In my reading, I found a large number of grammatical and typographical errors as well as confusing and poorly worded sentences.

Reviewer 2 implied that your measure of Frequency of Occurrence was inconsistent with their expectation. I tried to follow up your reference to Pinkas 1971 to check the definitions, but was unable to do so. I could not find a journal called Fisheries Bulletin. There is one called Fishery Bulletin published by NOAA, but the volume numbers do not match your citation. I found some articles by a fisheries scientist from California named L Pinkas but they did not match your title or journal. I did check Web of Science and Google Scholar but not spend a huge amount of time on this search, so I might have missed something.
• Please check that your reference is correct and complete.
• Confirm that the article contains the information for which you cite it, and specifically that the definition agrees with yours. In general, if you have not read the original article but taken the information from a different article, you should refer to the original source and add ‘cited by . . .’ giving the secondary source.
• If the article is obscure, many readers will not be able to find it, so consider an alternative source for your method unless it is the original source that created these terms. In that case, you might also cite a more recent clear example of the usage.
• Include the equations defining each of your measures in your methods so that readers do not have to search for the meaning of the terms.
• As the reviewer notes, it is still not clear how you estimated biomass for unidentified taxa.
• I think some mention of how you avoided double counting of remains would also be helpful.
• As with the reviewer, I had expected that Frequency of Occurrence would refer to the number of pellets containing a particular prey type, not the proportion of items. We could be wrong in this assumption, but you should confirm that the definition you used matches that widely used by other studies of animal diets. Do not just check the burrowing owl literature on this in case an author in this field made an error that was repeated by subsequent authors. If you wish to use a different definition of the same term, reader confusion may arise, so you must justify this and make your definition completely clear.

With regard to Table 1, I agree that biomass should be included, although perhaps it is included as Volume. It is critical not to change terms as this leads to reader confusion. Be sure that you clearly indicate when referring to it that this is estimated biomass because it was not directly measured.
• It would be helpful to provide sub-totals and final totals within the table for the categories that you refer to in the text. You could do this as additional rows, for example for different rodent families, all mammals, and all vertebrates. This will allow readers to confirm more easily your statements about the relative contributions to the diet of different taxa.
• I have provided an alternative heading for the table so that your abbreviations are referred to directly, not as footnotes requiring scrolling down.
• Within a column, you should provide a consistent level of precision, for example, not switching to whole numbers (25) instead of decimals (25.00). For very small contributions, you can write < 0.01 in a column where you use two decimals for other taxa.
• I found some spelling errors and inconsistent plurals in the table. Please check all taxon names very carefully.
• Be sure that you use consistent spacing between groups of higher taxa (no space before Uropygi).

Language – I have provided a pdf indicating many problems, including incorrect capitals, inconsistent plurals, and awkward sentences. When you have finished your revisions, please re-read the manuscript with great care checking for consistent tense use and capitalization. Then arrange to have a fluent English writer carefully read the entire manuscript to check the grammar and help you improve awkward or confusing sentences. If errors persist, the manuscript will have to be returned for another round of revisions, unnecessarily wasting my time and yours.

Other comments
Fossorial – The reviewer indicates that Burrowing Owls are not fossorial. Based on the rodent literature, my understanding is that fossorial applies to species that spend most of their lives underground in burrows. Small mammals that use burrows for sleeping, food storage, and parental care but spend substantial time outside foraging and mating are usually referred to as semi-fossorial. It looks to me that the literature may not be completely consistent in this regard. Please find a reliable source for the terminology so that you can be sure you are using it correctly.

Diurnal – There is inconsistency between L33 and L180 as to whether the owls are diurnal or crepuscular. Again please find reliable definitions of these terms and apply them correctly. If they don’t match what burrowing owls do, it is more important that the readers understand the owls than that a label is applied.

Chi square – In the Methods, you describe the chi-square as a test of association, which seems appropriate. In the Results (L166 and elsewhere), you describe it as a correlation. I am not sure you can have a correlation with a category. Please check that this term is valid as you used it and change, if necessary.

If anything is unclear about my comments or those of the reviewers or you are unsure how to proceed, please contact me directly by email (donald.kramer@mcgill.ca) so that we can avoid another round of revisions.

Reviewer 1 ·

Basic reporting

Reporting is clear and unambiguous.

Experimental design

The research question is adequately defined and its analysis well designed.

Validity of the findings

OK the results are meaningful.

Additional comments

No comments fro the authors

·

Basic reporting

No comment

Experimental design

No comment

Validity of the findings

No comment

Additional comments

Dear authors, I believe that in this new version of the manuscript most of the questions that I pointed out in the first review were resolved and the changes made resulted in an improvement in the manuscript. However, I still have some suggestions for the manuscript and some issues that I think are still unclear in relation to biomass assessment, which I have pointed out below.
Sincerely.

Felipe Zilio


INTRODUCTION
In the first paragraph, I’d like suggest a few changes about distribuiton and movements of the Burrowing Owl (BOWL). – “The North America races of Burrowing Owl (Athene cunicularia Molina 1782) are distributed from southwest Canada, through the west and central USA (but also in Florida) and Mexico, although most of the northern populations migrate to southern USA and Mexico (Marks et al. 1999).”
Lines 29-30 – “... and a great number of individuals spend part of their yearly life cycle in the three countries some portion of the year...” – This might be deleted
Line 33 – “The Burrowing Owl has diurnal activity, although hunting mainly at dawn and dusk (Coulombe 1971). - I understand your point, but BOWL is not a fossorial species. It uses burrows to make nests and protect itself from predators, as you mentioned later.

STUDY AREA AND METHODOS
Pellet Collection and Analyses
Line 107 – “We collected pellets at active burrows located along 20 random transects of 1 km x 200 meters where selected at random and each one measured 1 km long by 200 m wide...” – you repeated the information about transects.

Lines 116-117 – About FRO.
Now I understand what you meant when you referred to the Frequency of Occurrence. I usually refer to the Frequency of occurrence as the percentage of pellets that contain a certain type of prey. The total of prey items (or individuals) found in the sample I know as a Relative Density (or Relative Frequency), so I was confused when you presented your results using phrases like “Invertebrates represented 90% of prey items consumed…” or “Insects represented 84% of the items in the diet…”. The raw data also helped me to understand what you meant by FRO. So, many of my comments related to this subject (biomass measurement, etc.) in the previous review no longer make sense. However, although it is clear to me now, I still think that using Frequency of Occurrence or FRO instead of Relative Density might confuse and induce readers to misinterpret their results.

Satatistical analyses
Lines 147 – “Average anual precipitation is 427mm (INEGI, 2005) - If this is the average rainfall for Llanos La soledad, it should be in the Study Area section. Otherwise, I don't see why to bring this information here, since on lines 163-165 you showed the average rainfall over the years sampled.

RESULTS
Lines 157-159 – “Insects, primarily from the Orders Coleoptera (IRI=41%; 54%) and Orthoptera (IRI=10%; 26%), represented 84% of consumed items but contributed only to 12% of the biomass (Table 1).” - the second value refers to FRO? Table 1 does not show Biomass data.
Lines 166 – “There was a highly significant and small correlation between winters and prey items...” – I suggest: “There was a highly significant, but small correlation between winters...”
Lines 177 – “...were very stable among years with a relative biomass between 11 and 13% (Table 1).” - Did you delete the biomass information from Table 1 or use biomass as Volume? If you excluded, I think you should include biomass in Table 1. In the first review, I asked how you estimated the biomass of unidentified birds and I still don't understand. On lines 123 to 128, you explained how you estimated biomass for idenfied species, but not for unidentified species. Unless you measured bones or feathers and inferred the biomass of each bird, then calculated the total biomass of unidentified birds for each year, how did you estimate the biomass of a specimen you don't know what it is using a reference collection of a museum?
Table 1 – You should include the Biomass data and I suggest changing the P (presence) for N (Number of itens or individuals).

DISCUSSION
Lines 236-241 – “According to IRI (Pinkas et al. 1971), the main food component was invertebrates (IRI = 98%; 90%) Insects (IRI = 97%: 96%) with the highest Coleoptera consumption (IRI = 41%; 54%) and Orthoptera (IRI = 10%; 26%). Now, there are either large prey that are eaten infrequently or predominate in the samples because they are slowly digested such as vertebrates (IRI = 1.7%; 10%) and Arachnida (IRI = 0.44%; 2%), as mentioned by Hart. et al. (2002) (Table 1).” - I think that the IRI data are results of your study, so they should be in the Results, not in the Discussion. Here, just present the name of the taxon you are referring to. The reader, if necessary, will go to the Table or the results to check the values.

---

## Round 0.3 · Minor Revisions

I am returning your manuscript to you without a complete review. The reason for this my preliminary inspection of your rebuttal letter and version 2 manuscript indicates that you have paid insufficient attention to important points that the reviewer and I mentioned. Therefore, the manuscript remains unpublishable.

It is critical that you carefully address each point raised and that your rebuttal specifically states what changes you have made. It is not enough to state that you addressed the problem. You must detail what response you made and be sure that you clearly address all aspects.

For example, I spent some time attempting to verify your measure of frequency of occurrence, and I detailed in my previous decision how I was unable to do so because the journal listed does not seem to exist. Your rebuttal letter only mentions that a new revision was made and references cited. You should be informing me what the correct reference is and that you confirmed that it contained the information stated. I had to check the manuscript to see what you had done, and I found that you had not made the required corrections. The reference in question still remains in the literature cited (Pinkas 1971) and it is different from the citation in the text (Pinkas et al 1971). Following my instructions as to how to address a secondary source, you provide an additional reference to an article citing this. It is an article in Spanish in a relatively obscure journal, unlikely to be widely available to readers. It appears that you made no effort to determine whether the reference is correct or in error and whether the authors have correctly reported what the author(s) said. If they had the reference wrong, they may also have had the information wrong. Repeating published errors is poor scholarship that weakens scientific communication. I would have expected you to verify the reference and the information or find an alternative source. For a topic as general as diet frequency of occurrence, it should not be necessary to cite a secondary source in such a journal.

The reviewer and I raised concerns about your measure of frequency of occurrence. Your response indicated that other authors had used the term but did not address the specific formula to show that their definition was the same as yours. Furthermore, your response did not include the references for the authors you cited so I was unable to verify. However, one reference (Hall et al. 2009) was included in your manuscript, so I checked that. These authors used a definition equivalent to that of the reviewer and different from your FRO (L121-123). The way to avoid such confusion is to include the formulas with carefully defined terms. Although your rebuttal states that you added the formulas, you have done so only completely. All terms need to be defined in a logical order. By starting with IRI and then defining its components, you fail to clarify the components. Please seek advice from colleagues or published works to see how to present such information unambiguously.

Finally, I asked that after revision, you find a native English speaker to carefully review the entire manuscript. Your rebuttal does not state that you did this and the evidence of failure to do so is also in the manuscript. There are fewer errors than last time, but there are still too many.

Please return to your revisions and make sure that all points have been covered. Revise your rebuttal letter to make sure that your answers are clear and complete before resubmitting the manuscript. Note that I have only looked at a few issues and not completely evaluated your revision due to its incomplete nature.

---

## Round 0.4 · Major Revisions

Unfortunately, I must return your manuscript again without a complete reading. Your response to my previous comments is not satisfactory.

First, your rebuttal improves the response to my comments on version 1 as requested but fails to include any response to my comments on version 2.

Second, I checked quickly to see how the confusion about frequency of occurrence was resolved. Your definition continues to be based on the number of individuals of a particular prey divided by the total number of individuals of all prey types (L124-125). You now cite valid references to support this definition, and I appreciate the links which made it easy for me to follow up. I checked two of these and found that they do not support your definition.

Woodin et al. (p. 10) have a measure called ‘diet’ which is equivalent to your measure. However, they also have a second measure called ‘percent-of-occurrence’ which they define as the proportion of pellets in which each food type is found. They present the ‘diet’ data in Table 11. They present the occurrence data in Table 12 in which they change the term to ‘frequency of occurrence’. Thus, your definition of frequency of occurrence does not correspond to that used by Woodin et al.

Hall et al. (p. 3-4) define ‘percent frequency of occurrence’ as the number of samples with a given taxon divided by the total number of samples. A sample is defined as all the pellets and prey remains collected from a given burrow on a given date. Therefore, Hall et al. use the term to refer to what the reviewer and I consider as normal frequency of occurrence measures and not to the one you used.
I am afraid you have somehow misunderstood the references, and that your measures are incorrect. Therefore, you will have to redo your analyses. This is a major revision.

As noted before, your presentation of the calculations remains not logically developed, partially redundant and hard to understand. As suggested, you should look at how similar information is presented in well-established journals (not just burrowing owl sources).

Third, my quick scan of a limited section of your manuscript found continued grammatical errors. For example, there are 4 errors between L121-127. I therefore suspect that you have not adequately corrected the English as requested. This frequency of errors would require that I return the manuscript again, even if there were no other mistakes, so it is better to address the language issue before resubmitting.

---

## Round 0.5 · Major Revisions

Because of repeated problems regarding the definitions of your prey measures, I initiated an email exchange to allow us to go back and forth to resolve misunderstandings and errors. I asked you to send me the sources of the definitions and an Excel sheet showing your calculations, which you did. I reviewed this information and sent you back comments and an Excel sheet indicating places where your wording conveyed incorrect information and one measure that I felt was incorrectly calculated. I also noted that you should not simply take my analysis as authoritative but should be certain that I am right or show where I am wrong. In the most recent email, you did not specifically state that my suggestions were correct but did indicate that you were revising your calculations, which I assume means that you agree. Below is the main text of my email. I am unable to add the Excel sheet to my submission. Before re-submitting, please consult with an English-speaking scientist to be sure that your Methods accurately describe your calculations and that those calculations argre with the literature. Where there is not a literature precedent (perhaps FO calculations for groups of taxa), you will have to describe in more detail how you did the calculations.

1. Frequency of Occurrence (FO). When I look at the Excel sheet, it is clear that you calculated FO for each species by dividing the number of pellets that contained the species (numerator) by the number of pellets examined (denominator) and multiplying by 100 to convert to a percentage. This agrees with the references you provided, although neither author is explicit about converting the proportion to a percentage except for Marti et al when referring to the IRI calculation. Your Excel calculation does not agree with the wording in the methods of your manuscript. You wrote “The Frequency of Occurrence divided the number of prey consumed per pellet by the total number of pellets per season”. ‘Number of prey per pellet’ implies the mean value (number of prey divided by number of pellets). For example, following your manuscript methods for the Acrididae in the first year, you would have divided 70 acridids by 66 pellets (= 1.06) to get prey per pellet and then divided that result by 125 pellets examined (= 0.008). The correct Frequency of Occurrence is 66/125 = 0.528 (or 52.8%). Therefore, the description of the formula is what needs correction, not the calculation. That description must include the conversion to percent so that readers are clear what the values they see mean.

2. FO for taxa above the species level. In the Excel sheet, I did detect a problem. When you calculate the Frequency of Occurrence for aggregate categories such as genera or families, etc., you add the FO for each component. For example, FO of the Cricetidae is the sum of the FOs of each of the three species and the FO of Mammals is the sum of the FOs of the four mammalian families. This will lead to incorrect (higher) FO values because it ignores the possibility that some pellets will contain more than one species. I set up a new page with a hypothetical simplified example in your Excel file (titled ‘Example’) to illustrate this. In this example, there are two beetle species and one mammal in the pellets. The two beetles have FOs of 90% and 30%. However, they sometimes co-occur in the same pellet. I believe that the correct FO for invertebrates is 90% (column F), not 90 + 30 = 120%. This will affect your calculations of IRI.

3. Numerical percentage. The second measure you used is what Marti et al. and Munoz Pedreros et al. refer to as ‘Percentage occurrence by number’ and Santana et al. call ‘Numerical percentage’. This is the value obtained by dividing the total number of items of each taxon by the total number of items of all taxa in all pellets and then multiplying by 100. Your calculation is valid but your terminology is inconsistent and confusing. In the Excel sheet, this is called Frequency of Occurrence by Number of Prey. In the heading to Table 1 in your manuscript, you call it simply ‘Number of items (N)’. This is incorrect. To use the same example as above, the number of items for acridids in the first year is 70, but column N shows 23.1 acridids (70/303*100) which is value for Frequency of occurrence by number in your Excel sheet. In the methods of your manuscript (L122-123), you seem to call this Frequency of Consumption and use N to refer to the total number of prey of each taxon, not the percent of total numbers. Thus, there is inconsistency in terms, abbreviations and description of the value.

4. Biomass percentage. The biomass calculation is also unclear. The percent biomass should be the estimated as total biomass of each prey taxon divided by the total estimated biomass of all prey taxa combined, multiplied by 100. I cannot judge the accuracy of your measure because the Excel table you sent simply included the percent biomass of each taxon and not the value of the biomass per item or the biomass total of each taxon. The manuscript incorrectly describes this calculation on L.134. The calculation would be biomass of each item times the total number of items, not the frequency of occurrence. On L121, you call this Percentage of the biomass. In the heading for Table 1, you call it Estimated biomass consumed.

5. Index of Relative Importance (IRI). IRI is described by Marti et al. and Santana et al. as the sum of the numerical and volume percentages multiplied by the percent frequency of occurrence. However, you describe it (L125-126) as the sum of the number of prey and the percent biomass multiplied by the frequency of occurrence. It should be numerical percentage not number of prey. Again, the written Methods seems to be the problem because the IRI is correctly calculated in the Excel sheet you sent and the value agrees with Table 1 in your manuscript. Table 1 also provides another measure IRI%, which may be useful but was not defined or even mentioned in the caption. Although the calculation appears to be correct, the error in frequency of occurrence for grouped taxa will result in an error in this calculation.

6. There also appear to be some errors in the Excel sheet. I noticed that FO for sciurids in the first season includes unidentified rodents. However, this error does not seem to appear in the manuscript Table 1, so I don’t know how to explain this. Please be sure there are no other errors in the actual calculations.

My advice is that you check to confirm that my interpretations are correct. Although I have done diet studies, I was not previously familiar with all these measures, and I am not a specialized expert. As authors of the manuscript, you are taking responsibility for its validity, so it is important that you agree with my points not simply accept them on my authority. It is important that you choose clear terms for each of your measures, define them mathematically at first use including conversion to percentages, define the symbols to be used at the same time, and then use them consistently throughout the manuscript. To me, frequency of occurrence, numerical percentage, biomass percentage, index of relative importance, and index of relative importance percentage, seem to be least ambiguous. When you come to define the formula for IRI, use the symbols previously defined in your manuscript and describe how you replaced volume by biomass in your study. Using symbols from the literature that are not in your Methods just adds confusion for readers.

---

## Round 0.6 · Minor Revisions

Your description of the formulas used to analyze diet now seem to be correct. I therefore undertook a more thorough review of the text, something I had not done when it seemed that your calculations could be in error.

Your manuscript is now much closer to being ready for publication. However, numerous small errors remain. Many are grammatical, including agreement of subject with noun, changes of tense, and use articles ‘a’ and ‘the’. I am surprised that so many errors remain after submitting the manuscript for editing in English and that there are few if any grammatical corrections in the track-changes version. Is it possible that the changes proposed were not included in the final document? I will provide an annotated pdf with problem words or punctuation indicated by yellow highlights with an inserted comment to provide directions (such as delete) or an alternate wording that seems clearer or more concise to me.

There are several more substantial issues which I detail below. I hope that my comments are clear enough. If you have any doubts about my meaning, please email me directly for clarification. I would very much like this to be the final revision. Please check the English carefully; if there are errors in the new text, I will have to return it to you again. The changes are not numerous enough to justify an editing service. If you wish, you can send me the revised wording to check before submission (not the whole manuscript, just new text that you had to address some of the questions).

1) Biomass. L14 and many other places. In biology in English, biomass usually refers to the mass of living organisms in a particular area or ecosystem. I don’t recall seeing it used for the weight of diet items. I suggest using either mass or weight, with the abbreviations M or W. Technically, mass is the correct term, but weight is widely used in the literature and will be well understood. After you decide which term to use, please use a search tool to be sure you find and change every mention, including the figures and tables and their captions or headings.

2) L22, 173. I did not see statistical support for lower niche breadth in season 2. In Fig. 2, the CIs overlap extensively with the other years.

3) L62, 65, 67, 69. Because there is a large difference in percentages between numerical, weight and possibly frequency measures, it is important that you indicate clearly what measure you are referring to in your review of the literature. It is not necessaray to use the terms use define later; you can simply state something like ‘70% of the prey counted’ or ‘70% of the estimated volume of all prey’. Statements on Lines 72, 73 were clearer.

4) L118. Having indicated that you counted the items, you should explicitly state that you estimated their mass/weights. Then provide the information from L137-144 about how you estimated this.

5) L118. Identification references L131-135 belong here more logically.

6) L119. I think that more concise and clear terms would be ‘precent frequency of occurrence’, ‘numerical percent’, ‘percent weight’ and ‘percent index of relative importance’. If you agree carefully search for every mention, including tables, figures and their headings/captions to change all cases.

7) L119 These are taxonomic levels of prey, not trophic categories.

8) L124. Insert your measure of mass/weight here to go with frequency and numerical calculations.

9) L125, 126 Instead of V in the formula and then having to explain that you replaced volume with weigh, it would be better to use mass or weight (M or W) in the original formula. Also, use M% or W% to be consistent with the other components.

10) L152. It is not clear if ‘their overlap’ refers to niche breadth as implied by the sentence structure. If this is the meaning, I don’t see why another index is needed. I looked for a mention of Horn’s index in the Results to help me understand what you meant, but I could not find any mention. Please clarify and check the English carefully. Do not include methods that were not used in Results. If the index was used, please be more specific.

11) L151. Specify which diet variables were used for the niche breadth measures and why. In Results, it implies that frequency and mass were used, but it is not clear why you would not use number as well. Also, be clear whether you used the ‘raw’ or percent values. The text implies that percent values were not used, but the Methods indicated that this was the calculation used. I think it would be good to add a sentence stating that Smith’s FT ranges from 0 with 1 and what this means. I assumed that ‘overlap’ referred to overlap between years, but after a quick read of Smith 1982, it seems that overlap refers to availability. All this needs clarification so readers understand what you did and what it means.

12) L155-156. The sentence implies that the association examined was between frequency and biomass, which is not the case. I think you mean ‘To test for an association between years and the diet composition, we used χ² contingency tests (Zar 1998) followed by Cramer’s phi coefficient (ϕc, Cohen 1988) as a measure of effect size. For years, we used annual rainfall (reference). For diet, we used . . .’ [it is not clear here or in results what you used, but you need to specify all taxa and all measures (frequency, number, weight, and/or IRI percent) used in this comparison. Check the English carefully so there are no new errors.]

13) L165. You are referring to counts from the pellets, so you should refer to the pellets rather than ‘consumed’ which might have included items not identified.

14) L172 What these values represent is not clear. Why is there only value when you measured three years? Is it the three seasons combined? Why not include number? You should use the terms in your Methods. If you did not use the percent values, you should explain in methods. If you did use the percent values, they should be specified here.

15) L173. Why significantly smaller when CIs overlap?

16) L175. Check. I thought the rainfall in the second year was greater than the other two years, not less.

17) L177. Be specific about which measure you used. Was this frequency percent?

18) L180. It is confusing to readers when you switch order and terms for no good reason. On L177, you said ‘between winters and prey items’. No you say, ‘between biomass and years’. Readers would infer that you mean different things by ‘winter’ and ‘year’ when you do not. Changing the order just makes it hard to follow the comparisons.

19) L181. After giving values for invertebrates but not vertebrates. This is very incomplete and confusing!

20) L182. This is an incomplete comparison. You refer to greater consumption but not which year.

21) L183-184. You should provide common and scientific names at first mention in the manuscript and then only use one. I think common names will be easiest for most readers. Please check the entire manuscript to be sure that you always give the scientific name the first time you refer to a species and never repeat it.

22) L185-187. This sentence is out of place; you are referring to associations of years in Results, so should not return to niche breadth. (This could be a point raised in the Discussion.) In addition, the sentence is unclear because you don’t provide the values for the second year. You can’t expect readers to have memorized them! When you cite FO values, you seem not to be giving % as your methods indicated and you don’t say FO of what group.

23) L268. The last paragraph of the Discussion, by referring to future research needs, is more appropriate as a second paragraph in the Conclusions.

24) Figure 2. Please use larger fonts for the axis labels. Center the years on the tic marks on the x-axis. I think you should add (FT) to the y-axis label.

25) Supplementary data. Please include a key to identify all abbreviations.

---

## Round 0.7 · accepted · Accept

I am pleased to recommend this manuscript for acceptance. It has been a long process, but the article has been substantially improved over the numerous revisions.